# A Novel Model for Generating Creative, Community-Responsive Interventions to Reduce Gender-Based Violence on College Campuses

**DOI:** 10.3390/ijerph18157933

**Published:** 2021-07-27

**Authors:** Sophia Graham, Caroline Cao Zha, Abby C. King, Ann W. Banchoff, Clea Sarnquist, Michele Dauber, Michael Baiocchi

**Affiliations:** 1Department of Epidemiology and Population Health, Stanford University School of Medicine, Stanford University, Stanford, CA 94305, USA; sophiebgraham@gmail.com (S.G.); king@stanford.edu (A.C.K.); 2Department of Human Biology, Stanford University, Stanford, CA 94305, USA; caroline.zha212@gmail.com; 3Stanford Prevention Research Center, Department of Medicine, Stanford University School of Medicine, Stanford University, Stanford, CA 94305, USA; banchoff@stanford.edu; 4Department of Pediatrics, Stanford University School of Medicine, Stanford University, Stanford, CA 94305, USA; cleas@stanford.edu; 5Stanford Law School, Stanford University, Stanford, CA 94305, USA; mldauber@stanford.edu

**Keywords:** sexual assault, gender-based violence, community violence, college, campus, social ecological model, prevention, intervention

## Abstract

Currently, the most successful prevention interventions against sexual violence (SV) on United States college campuses target modifications at the individual and interpersonal levels. Community-level interventions have been under-developed for college campuses. To address this gap, we employ a citizen science model for understanding campus community factors affecting SV risk. The model, called *Our Voice*, starts by engaging groups of college students to collect data in their own communities, identifying factors they view as increasing the risk of SV. In facilitated meetings, participants then review and analyze their collective data and use it to generate actionable community-level solutions and advocate for them with local decision-makers. We share findings from a first-generation study of the *Our Voice* model applied to SV prevention on one college campus, and include recommendations for further research.

Sexual violence (SV) which, for purposes of this paper, encompasses sexual assault and harassment, is a serious public health issue. Female students on college and university campuses are especially vulnerable to experiencing SV, and evidence suggests that transsexual and non-binary students are also at high risk [1,2]. Various interventions have been implemented to reduce SV within college settings; however, there is a dearth of community-level SV prevention programs to complement existing interventions. 

The purpose of this study was to adapt and pilot an existing framework for citizen science research—*Our Voice*—for use in the development of campus-level interventions to reduce SV. Given the sensitivities associated with SV, there was a need for adaptation and assessment of acceptability. This piloting involved modifying the protocols that have been used previously with *Our Voice*. We report the full results obtained from this first-generation investigation of the model and suggest directions for future research. 

In the *Our Voice* model, community members document features of their local environments that help or hinder their ability to stay safe and healthy. They then meet to review their own data and collectively generate locally-relevant actionable insights that they can share with decision makers to advance change. The model has been applied to generate community-driven solutions around a range of health issues in communities around the world [3,4,5,6,7]. 

As a research model, *Our Voice* represents a novel model to the college SV prevention field by including the following suite of features: (a) it is neutral with respect to the advocacy and interventional steps taken by participants; (b) it contemporaneously generates data from the experiences of the community itself; (c) it offers the researchers insights into the decision-making process of community members as they identify and prioritize issues, and as they discuss potential solutions; (d) it produces several potential interventions that community members identify as meaningful; and (e) after selecting particular interventions to move forward with, it scaffolds a process for participants to advocate for and implement interventions generated and conducted by the community itself. 

In this article, we first review the research on SV on college campuses, current theories of interventions to prevent it, and the need for community-level prevention interventions. We then introduce the *Our Voice* model, laying out a theory for applying it to college SV prevention. We share initial results from a pilot study and discuss the model’s potential value in addressing SV at the campus level. Finally, we discuss lessons learned from the pilot study and outline our recommendations for further research.

## 1. Introduction

In this section, we first review the research on SV on college campuses. We then examine current SV prevention interventions, and identify the need for community-level programs to prevent SV. We provide two examples of community-level interventions working to reduce violence within local communities. Finally, we introduce the *Our Voice* model, laying out a theory for applying this citizen science framework to college SV prevention.

### 1.1. Sexual Violence (SV) Background

The Centers for Disease Control and Prevention (CDC) notes that in the United States in 2015, 43.6% of women (around 52.2 million women) reported experiencing some form of contact sexual violence in their lifetime, while one in five women (21.3%; approximately 25.5 million women) experienced a completed or attempted rape in their lifetime [8]. Women who have experienced SV may have immediate as well as lasting physical and mental health impacts, such as injuries from the assault, chronic health conditions, and post-traumatic stress disorder symptoms and sequelae [9,10]. Additionally, SV has a large economic impact. For example, victims of SV use more medical services in the years following an assault than women who do not have a history of SV [9]. The long-term economic impact for a victim of rape is estimated in the U.S. at $122,461, which translates into a total economic burden of $3.1 trillion in the U.S. [11].

College women are particularly vulnerable to SV. Studies consistently document the prevalence of sexual assault among college women, with approximately one in four U.S. women experiencing assault during their four years in college [1,2,12,13]. SV risk is not equally distributed, either between campuses or throughout a woman’s time in college. Between campuses, variations in campus demographics and culture may lead to differential risk for SV; for instance, campuses with strong Greek life or athletic presences, as well as campus cultures that set the stage for binge drinking and/or casual sexual hookups, have been associated with increased likelihood of SV [2]. Within an individual’s college experience, risk for SV is also differentially distributed. Many studies have indicated that the risk for SV is highest during freshman year, particularly during fall term, when students may be the least familiar with campus and campus culture [1,2].

It is important to recognize the multifaceted and intersectional risk people face as potential victims of violence, as well as the various stakeholders that maintain or can support change in these structures [14,15]. The social ecological model explores the complex interplay between individual, interpersonal, community, and societal factors that contribute to an individual’s risk of experiencing SV [14,16,17,18]. The model provides a framework for identifying the array of different factors leading to SV. The Whole School Approach, as applied to college SV prevention in McMahon et al., further expands upon the multi-level interplay of the social ecological model by examining SV as a community-level issue that requires active engagement of all community stakeholders to reduce it [15].

### 1.2. The Need for Community-Level SV Prevention Programming

In this subsection, we first review current SV prevention programming, noting the focus on individual- and interpersonal-level prevention strategies. We then present the need for interventions that target community-level factors affecting one’s risk of experiencing SV—with these interventions complementing existing individual- and interpersonal-level programming.

#### 1.2.1. Preventing SV on U.S. College Campuses: Existing Research and Theory

College campuses in the United States are now required to offer campus-wide SV prevention programming [19]. In many cases, the programming deployed consists of brief online training which delivers legal and psychosocial information, but evidence is lacking that they reduce rates of SV [20]. Notably, risk reduction and empowerment self-defense interventions for women have strong evidence of success in reducing SV rates on college campuses. These interventions teach potential victims, most often women, to assess situations for risk factors and provide participants with verbal, physical, and emotional skills to avoid and resist SV [21,22,23]. However, these programs have not been implemented widely. Nearly all college campus SV interventions focus on individual-level and interpersonal-level factors [17,22,24,25,26,27]. The social ecological model suggests that community- and environmental-level factors also play a complementary role in reducing the risk of SV, yet few interventions targeting the community-level have been developed [17,26]. These community-level interventions, as described by DeGue et al., work to modify characteristics of the college campus that increase SV risk—“these include approaches that operate to change community-level norms, risk factors, and policies within communities” [25].

While some of the individual, interpersonal, and institutional interventions described above may be effective in reducing the risk of SV on college campuses, there is a lack of SV prevention programming that focuses on community-level change [16,25]. A report published by the CDC’s Division of Violence Prevention has emphasized this overall dearth in the U.S., calling for evidence-based interventions targeting at community level [25]. A review by Moylan and Javorka highlights the need for better understanding of SV risk at the community level [26]. They point to the wide variation of SV rates across campuses as an indicator of the effects of campus-level factors. Community-level factors that vary from campus to campus can foster different risks for SV [28]. Factors include, though are not limited to, campus policies (e.g., availability of resources for victims, the university’s alcohol policy), campus features (e.g., the presence of fraternities, the ratio of on- vs. off-campus residences), and campus cultures (e.g., a heavy alcohol culture, the normative views and status of women on campus) [26,29]. McMahon conducted a similar review, specifically analyzing campus-level factors that may foster bystander intervention and argues that sense of community, pro-social modeling, and positive social norms, policies, and physical environment encourage bystanders to intervene [30]. Research by Potter et al. also points to shifts in community social norms and pro-social modeling as important factors that may affect SV risk [31,32]. By intervening on the campus-level factors that can affect a SV-prone environment, this could in turn reduce the rate of SV on campus [24,26,33]. These connections of factors are referred to as the “the web of prevention” [23] (citing Hamby & Grych, 2013).

#### 1.2.2. Community-Level Interventions: An Overlapping Web of Prevention

As part of a web of prevention, community-level interventions offer complementary benefits to other types of interventions. Community-level interventions can create changes that affect interpersonal interactions without necessitating case-by-case interventions, changing behaviors without directly intervening on individuals. Additionally, because of the strong social desirability bias surrounding SV and a lack of readiness to change, perpetrators may be especially unlikely to willingly participate in a SV intervention [24]. This can make it difficult to reach the populations at highest risk for perpetrating SV. Community-level interventions sidestep this challenge by targeting community norms as well as physical and social contexts that affect the behavior of all community members. This has the potential to make an intervention’s effects more widespread than they otherwise might be. Further, some community-level interventions (e.g., changes in housing policies) do not require repeated education to the same degree as individual- and interpersonal-level interventions, potentially avoiding annually recurring costs. These arguments should not be understood as devaluing individual- or interpersonal-level interventions, but rather as identifying unique strengths which community-level interventions may offer as part of a web of prevention.

Thinking in terms of community-level interventions requires a shift in emphasis from the types of individual- and interpersonal-level interventions typically deployed on campuses. Ideally, community-level interventions engage community members throughout the intervention development and deployment process. Fostering change within a community requires the recruitment, activation, and continual engagement of community stakeholders [17]. Including community members can help researchers better understand the complex social ecological interplay affecting SV risk for college students. Actively involving these groups can also result in interventions that fit the broader community’s needs, starting with some level of buy-in. For example, interventions solely created by adults without adolescent involvement may reduce their efficacy or sustainability [34]. Additionally, actively involving and centering historically minoritized or marginalized students (e.g., students of color, those from economically impoverished circumstances, non-heterosexual or non-cisgender—i.e., LGBTQ—students, intersections of historically minoritized or marginalized identities) will broaden the types of interventions produced to better target the wide array of students on campus [35].

Casey and Lindhorst argue for a multi-level ecological approach that includes direct community engagement to better identify specific targets for change of particular relevance to that community [17]. Likewise, the CDC highlights the importance of including community members in problem identification, relevant data collection, and solution-building to create sustained effects [36]. Furthermore, McMahon et al. assert that engaging the various community stakeholders and placing students at the core of the process are essential to creating a comprehensive, community-level prevention approach [15].

The citizen science model applied in this study—*Our Voice*—can help address this need for community-level, community-driven SV interventions, as laid out in this section. *Our Voice* calls for active participation and ownership of community members in all aspects of the research process—problem statement, data collection, analysis, and operationalizing insights from the research. Citizen science and *Our Voice* are described in more detail in the next section (Section 1.3).

#### 1.2.3. Examples of Community-Level Violence Prevention Programs

There has been some exploration of community-level programs to reduce SV. A program targeted at reducing intimate partner violence (IPV) and HIV risk implemented in Kampala, Uganda, entitled SASA!, found that the community-level program was associated with a reduction in rates of SV [37,38]. In the SASA! intervention, community activists with an interest in the topic attending training regarding the concepts of gender inequality and power structures. These community activists were then motivated to share their awareness within their networks, through training, SASA! materials, and/or community events, which in turn led to these community members becoming empowered to disseminate the information to their own networks. All community members were encouraged to act on the learned concepts by changing gendered behaviors. The SASA! community mobilization method was associated with a reduction in the social acceptance of, and the rates of, IPV [37]. No such community-level intervention has been applied to college campuses.

Outside of the context of SV, there is evidence that community-level interventions can reduce violence. For example, the Institute for Community Peace (ICP) led a community-level intervention aimed at addressing the root causes of community violence in eight underserved U.S. communities [39]. Participants were given the opportunity to assess, identify, and address specific issues within the community (e.g., boarding up abandoned buildings, disrupting drug trade hotspots). This participatory approach fostered connections with other community members, which increased their sense of community efficacy and cohesion. Participants also discussed factors such as existing gender, class, and racial biases, which were the ultimate drivers of the issues they identified, and began creating community-based solutions to these concerns [39]. In addition, many individuals began implementing solutions discussed by the community in their own lives [39].

ICP’s work with community violence prevention suggests that community-based empowerment and primary prevention interventions may be an effective tool for reducing rates of interpersonal violence. Given the dearth of community-level SV interventions on college campuses, we introduce a model to search for beneficial interventions. In the remainder of this article, we describe a community-engaged citizen science model for generating and applying community-level interventions for SV prevention.

### 1.3. Introduction to Citizen Science—The Our Voice Initiative

In this section, we introduce an evidence-based model for generating community-level, community-driven interventions, called *Our Voice*. We then compare *Our Voice* to other existing models to highlight the usefulness of applying the *Our Voice* model to community-level SV prevention. We also expand upon the previous applications of the *Our Voice* model, a model which has been applied to generate community-driven solutions around a range of health issues in communities around the world [3,4,5,6,7].

#### 1.3.1. The Citizen Science Approach

The citizen science approach described in this article—*Our Voice*—is aligned with community-based participatory research (CBPR) and participatory action research (PAR) [40,41]. While this type of citizen science and these other approaches often share the goal of integrating community perspectives throughout the research process, and empowering participants, there are several important distinctions. Following the argument of Rosas et al. and King et al., citizen science approaches typically educate community members themselves on employing systematic and rigorous forms of data collection—enhancing their ability to collect and evaluate the data [40,41]. In contrast, other forms of PAR may focus on community organizations and similar stakeholders or “gatekeepers” in gathering information about the community, or use less systematic data collection formats (e.g., focus groups) for gathering resident perspectives or insights. The citizen science approach thus offers a model for engaging directly with community members on the individual level, outside of formal community–university partnerships. This type of democratizing format can enhance the flexibility of participation models. It can also spur feelings of empowerment among residents independent of participation by community-based organizations. It allows for the flexible creation of different groups (e.g., a cohort of women, a cohort of students who participate in Greek life) to help understand different subgroups’ perspectives, and possibly how heterogenous groups interact (e.g., through involving a cohort of mixed genders and mixed campus affiliations). Finally, many forms of citizen science have been particularly concerned with the social–environmental contexts and their impacts on human and/or planetary health and wellbeing [42,43].

The emphasis in citizen science “by the people” on greater levels of involvement in data collection, as well as analysis and interpretation, provides the opportunity for community members to identify, systematically collect, analyze, and utilize data to drive changes in their own environments that are meaningful and relevant to them. This is relevant for promoting equity within the community, as the influence of social and physical environments on inequities is often only noticed by those for whom the situation is unfair and unjust, and thus may be less well understood by those in control of a community’s decision-making levers or channels.

In addition to the important ethical principles generally recognized for implementing research in communities, there are particular dynamics involved with citizen science. The European Citizen Science Association (ECSA), led by the Natural History Museum of London, set out several principles for citizen science [44]. We highlight the following principles from their list: citizen science projects actively involve citizens in scientific endeavor that generates new knowledge or understanding. Citizens may act as contributors, collaborators, or as project leader and have a meaningful role in the project. Citizen science projects have genuine science outcomes. Both the professional scientists and the citizen scientists benefit from taking part. Citizen science is considered as a research approach like any other, with limitations and biases that should be considered and controlled for. The leaders of citizen science projects take into consideration legal and ethical issues surrounding copyright, intellectual property, data sharing agreements, confidentiality, attribution, and the environmental impact of any activities.

#### 1.3.2. Our Voice: Key Features

The *Our Voice* model relies upon a “citizen science” model to generate community-level data of community members lived experiences, aimed at widening the intervention lens beyond the individual and interpersonal levels [45,46]. This specific citizen science model, which fits within the community-based participatory research field, defines “citizen” as inhabitants of a particular locale or members of particular groups, without regard to immigration status.

The *Our Voice* model involves active engagement of members of the community in all aspects of the broader research process, including project planning, collecting, analyzing, and interpreting data, and generating data-driven change within their community [47,48]. To date, *Our Voice* projects have been implemented in over 20 countries, to address a range of health and social issues across diverse groups of community members [3,4,5,6,7,49,50,51]. This article represents the first application of the *Our Voice* model to drive community-level SV prevention.

Given the novel application of *Our Voice* to the SV prevention field, before describing the methods of this particular study, we will introduce the four steps that generally comprise the *Our Voice* theoretical model [47,48]:
**Planning and recruitment:** In a train-the-trainer model, the Stanford-based *Our Voice* research team equips and supports community-based researchers and/or other partners to facilitate the citizen science process. The community facilitators, who may include lay community leaders, researchers, or representatives from community-based organizations, determine both the general focus for the *Our Voice* project and the specific data-collection questions of interest. Once the topic has been determined, the community facilitators use flyers, email, word of mouth, and/or other relevant strategies to recruit participants for the collection, analysis, and use of their own data to advocate for meaningful and sustainable change within their community.**Collecting data while moving through the community:** Community facilitators introduce the project, train participants on data collection, and support participants in collecting data while moving through their community. Participants collect data using the Stanford Discovery Tool (SDT), a smartphone-based app available on both Apple and Android platforms. As participants move around their community, the SDT allows them to take geo-tagged photos and record audio or text narratives about the local environmental factors that can positively or negatively impact the specific issue under investigation [49]. Project facilitators provide participants with specific questions/prompts to encourage the data collection. These questions are usually worded broadly to allow for individual interpretation and to encourage participants to collect a wide range of data [49].**Priority setting:** Led by the community facilitator(s), participants together review the data they collect and discuss any common themes or trends they notice. They are then supported in building consensus around the identification and prioritization of specific challenges and opportunities in their community; brainstorming potential solutions and strategies; and deciding as a group which solutions they believe would be the most feasible and/or effective to pursue.**Group action:** Community facilitators guide participants through a discussion of how to put these potential solutions and strategies into practice, with emphasis placed on managing scope and increasing participants’ knowledge and skills for successful advocacy. Depending on the scale of the solution and the prior experience of participants, this may range anywhere from finding contact information for local officials to discussing how to effectively and persuasively present data in order to lobby for desired changes. During this phase, participants meet with relevant local stakeholders and decision-makers who can be instrumental in helping to address the group’s proposed solutions and strategies.

#### 1.3.3. Our Voice: Comparison with Existing Methods

The *Our Voice* model shares some features with two related forms of data collection—ethnographic fieldwork and focus groups. However, it departs from these two methods in critical ways that merit exploration.

Some of the insights generated by the *Our Voice* model could be captured by careful ethnographic fieldwork, in which researchers observe and describe communities without interfering, working to accurately capture and reproduce thick descriptions of a social dynamic under study [52,53]. In contrast to this kind of qualitative research, however, in the *Our Voice* model the process for studying the social dynamic is intentionally co-created by members of the community and researchers—including the targeting of a specific social dynamic, the formation of questions, and the recruitment of participants. Distinct from ethnographic fieldwork, the *Our Voice* model has modifiable design choices—e.g., recruiting either a heterogeneous or homogenous participant group can give divergent insights about both the targeted social dynamic, as well as potential solutions. This is an important feature of *Our Voice*—researchers can and should explore modifications of the design features discussed in the previous section (e.g., group size, group composition, guiding questions) so as to explore potential variations in outcomes.

In some ways, *Our Voice* is also similar to focus group methodology, in which community members are brought together and put in conversation in order to elicit varying opinions and experiences [54]. In both approaches, the interplay among participants can elicit more complex understandings of social norms regarding the topic under investigation. In focus groups participants recall experiences and feelings; however, *Our Voice* supports participants in systematically collecting data from their lived experiences in their community, in situ, using the SDT mobile app. As described in Step 2 above, the participants are asked to identify and record their experiences (visual, audio, etc.) over a specified period of time. These data are then automatically aggregated on the SDT Data Portal, and then distributed to participants in preparation for analysis and use during Steps 3 and 4. While focus group data are typically coded and analyzed by professional researchers, in *Our Voice* projects the data are generated, interpreted, and used by the participants themselves, based on their own experiences in the community. Contemporaneously collected data by community members themselves are particularly important for members of marginalized communities, who traditionally have had their concerns overlooked or misunderstood.

In the setting of sexual violence prevention, the *Our Voice* model is similar to the Photovoice approach [55,56]. *Our Voice* generally builds on the theories and best practices of Photovoice, with a shift in some notable ways. On the technical level, *Our Voice* uses a smartphone app to collect location information, geo-tagged photos and video, voice or text annotations, as well as simple ratings of local features as positive, negative, or both—extending the media collected, which can be integrated for later analysis. *Our Voice* also places the emphasis on resident-driven community-level solution building, taking collective action, and shared decision-making among residents themselves as well as with relevant community and research organizations. Several of the steps in the *Our Voice* model (described in Section 1.3.2) ask a cohort of participants to come together to jointly analyze their data, identify challenges faced by the community, propose solutions, and then collectively enact a realistic course of action. This is not a dramatic departure from Photovoice, but we want to highlight here two important shifts: (i) the enriched multi-component data and group-level analyses encourage community-level solutions and may improve community acceptance of the solution; and (ii) as part of this citizen science model, the participants play a more “front and center” role in the analysis and interpretation of the data. For prevention researchers, we note that the discussions focused on generating solutions, as well as the types of solutions the citizen scientists choose to pursue, are a particularly useful feature of *Our Voice* and similar community-engaged citizen science models.

#### 1.3.4. Our Voice: Previous Work

The *Our Voice* model has been applied as a community-based intervention to address a diverse range of community-level factors that create barriers to heathy living. In East Palo Alto, CA, for example, *Our Voice* participants identified unsafe and/or inaccessible sidewalk paths that impeded physical activity and used their findings to successfully advocate for the implementation of a comprehensive sidewalk inventory and repair [7,57]. In Anchorage, AK, LGBT older adults documented their community experiences that impacted feelings of social isolation and collectively generated ideas for improving social connectedness [3]. Of particular note, in a number of *Our Voice* projects that have been conducted around a range of community issues and in various locales, the increased feelings of empowerment and agency experienced by community members have led to positive ripple effects, as community members apply their newly acquired skills to other issues in their community [6].

In this pilot study, we extended the *Our Voice* community engagement model to the campus SV field in order to understand and inform the development of strategies to address the multitude of community-level factors that may affect rates of SV.

## 2. Materials and Methods

The primary aim of this study was to apply the *Our Voice* model to SV on U.S. college campuses. As noted earlier, the application of a research methodology focused on community-level SV factors is an underdeveloped approach within the college SV prevention field. The study took place at Stanford University located on the west coast of the United States and focused specifically on undergraduate women. The University’s *Our Voice* IRB document was modified to account for this pilot study and approved by the IRB (IRB #45330).

### 2.1. Study Design

A pilot study of the application of *Our Voice* to SV was conducted at Stanford University in spring 2019. Two student facilitators collaborated with the research team in developing data collection prompts that focused on the social spaces and situations on campus potentially associated with SV. An initial test application of the *Our Voice* methodology for SV in 2017 had suggested the importance of (1) using generalized prompts that did not bias data collection by directing students to consider certain spaces or types of danger, and (2) collecting data over a multiday period instead of during dedicated walks, as done in most prior *Our Voice* studies. Because community factors related to SV may not be encountered on a regular basis or within a single walk around the community, participants were asked to collect data for seven consecutive days while maintaining their daily routines. For example, the methodology test revealed that emphasizing unlit areas and hardscapes tended to elicit concerns about stranger assaults, which are uncommon on college campuses, rather than concerns about indoor spaces such as fraternity houses and dorm rooms where SV is more prevalent (see research by Steinmetz and Austin noting that college campus hardscape provoked high levels of fear among students [58]). Informed by feedback from this test, the student facilitators held two question-testing sessions with undergraduate self-identified women (*n* = 4 in each group) to assess the appropriateness of the data collection prompts. Participant feedback indicated that the prompts were clear, sensitive, and relevant. The final data collection prompts were as follows:
What are social spaces or situations at the university where you feel unsafe or uneasy?What are social spaces or situations at the university you feel uncomfortable in because of your gender?What are social spaces or situations at the university where you feel comfortable or at ease?What are social spaces or situations you think affect many of the university’s students’ feelings of discomfort or unease related to their gender?

### 2.2. Selecting the Target Population

Student facilitators suggested targeting potential student participants with previous experience in studying or advocating for SV prevention or other women’s health issues, in the belief that such selection criteria could help to form a group more interested in and/or dedicated to participating in the group action component of the project. Student facilitators also chose to exclusively target self-identifying women students given the high prevalence of SV against women.

Participants in the pilot study were recruited through emails to campus-wide mailing lists, particularly email listservs for students interested in women’s issues, SV prevention, and public service, as well as various dormitory-wide mailing lists. Interested students completed a Google Forms survey with contact information, class year, and campus community affiliation (e.g., student groups, affinity groups). Exclusion criteria included not self-identifying as a woman, not being an undergraduate student, previous participation in the question-testing pilots, and/or indicating ahead of time that they could not attend the meetings.

### 2.3. Data Collection Process

At the initial informational session, student facilitators taught recruited students how to download and sign into the SDT mobile app on their personal devices, and how to use the SDT to capture photos and audio or text narratives about their feelings or experiences of risk and safety at the university. The participants were provided with documents that contained a quick troubleshooting guide for the SDT, as well as all data collection prompts (see above) and instructions for data collection. Participants were given the opportunity to discuss and ask questions about the data collection prompts, as well as the project generally.

During the informational session, the student facilitators guided the participants in a “warm-up” exercise. Participants were asked to visualize a situation and then given a variation of the question prompts used for data collection. The student facilitators proposed this technique to encourage participants to collect data around specific, tangible experiences evoked during the data collection process (e.g., specific community-level factors), rather than noting generalized dynamics they or fellow students have faced on campus.

Participants collected data for seven consecutive days and received a daily text reminding them to collect data, but were not otherwise prompted by the student facilitators or researchers. Based on recommendations from the student facilitators, reminders were sent in the morning (between 7:30 am–10:30 am), and included a brief reminder to take photos throughout the day and accompany each photo with text or audio narratives describing why they had taken the photo. The timing of the reminders used in this study was strategically chosen to limit participants responding immediately, which could skew the type of data collected, and instead prime students to log experience when they occurred throughout their normal day.

Data collection guidelines were as follows:
Collect data for 15 min per day (approximately three to five photos and text/audio narratives per day).Only collect data about events that you experience/witness, not things you remember happening in the past or that you are told have happened to others.Refrain from changing your daily routine to collect data.Do not take photos of any faces or similar identifying features.If you do not encounter anything in your day that prompts data collection, upload a placeholder photo with the caption, “I did not feel unsafe or unusually aware of my gender today,” and, if relevant, explain why.

### 2.4. Data Analysis

The data analysis followed previous *Our Voice* protocols [3]. Data were uploaded for storage on the *Our Voice* secure server, and cleaned and transcribed (if in audio format) by research team members. Printed packets of the geo-located photos and accompanying text were assembled by the student facilitators. Student facilitators split participants into pairs to review the printed packets containing data collected by the group as a whole, and to discuss any themes they noticed [3]. In a subsequent larger-group discussion, each pair shared themes and sample data examples, recording their findings on large sheets of paper for group visualization purposes. Student facilitators guided the participants through organizing the data into major themes and developing a list of potentially actionable solutions. Due to time constraints, given the postponement of the study until near the end of the school year, the final “group action” step of the *Our Voice* process—in which members work with local stakeholders to implement solutions—was not seen to fruition in this first-generation study.

## 3. Results

### 3.1. Participants

A total of 18 undergraduate students responded to the initial email, eight of whom were excluded based on eligibility criteria (e.g., did not self-identify as a woman) or because they did not respond to follow-up emails confirming participation or did not attend the sessions. Ultimately, 10 undergraduate students—mostly freshmen and sophomores—were enrolled in the pilot, and attended the information session and the debrief session. Other *Our Voice* research suggests that group sizes do not need to be large in order to capture major issues and priorities for the community [3,47,48]. Various race/ethnicity groups were represented, including White, Latinx, and Asian. Participants self-identified as many different sexual orientations, including heterosexual, bisexual, and queer (we obscure exact numbers in each of the other descriptive categories for privacy reasons). Most of the participants lived in university-managed dormitory-style housing on campus, while several lived in on-campus houses primarily or entirely managed by student residents instead of university staff.

### 3.2. Stanford Discovery Tool (SDT) Data

Participants completed a total of 102 walks over the course of seven days during the University’s 2019 spring quarter session, using the SDT app to record 120 photos and 90 text/audio narratives of campus.

### 3.3. Priority Setting

Due to time constraints, the student participants met in a single two-hour group meeting the day after data collection ended to conduct the data analysis. Major themes identified by participants included Greek life, dorms and housing, academic and campus culture, personal space, direct interactions (e.g., time with friends, one-on-one conversations), passing interactions (e.g., social interactions while walking to class), large social events (i.e., parties or concerts), age differences (e.g., between graduate and undergraduate students), gyms, and heightened awareness or knowledge around SV (see Table 1 for examples). Participants noted both positive and negative aspects for most themes. Although participants were encouraged to ground data collection in direct experiences, some data did use recollections and generalizations, as shown in Table 1.

### 3.4. Solution Building

When all student dyads had shared their themes with the group, student facilitators prompted the participants to collaborate in ranking the issues, asking, “What are the most important issues that we’ve identified that we think need to be addressed, if any?” Based on this prompt, participants identified major themes that they believed were the most pressing. Once priority issues were identified, the student facilitators encouraged students to discuss potential solutions and strategies for addressing these issues.

Ultimately, participants prioritized four major themes—dorm bathrooms, campus cultural norms around male behavior, social spaces/events (e.g., parties), and classroom/academic gender expectations. According to student facilitators, the group chose these themes due to their actionability and relevance to the majority of participants. Within each of these major themes, participants collaboratively brainstormed SV prevention solutions and strategies, some of which could be community-led and others that would require administrator buy-in and engagement.

During this discussion, participants drew upon their past experiences working in SV prevention and/or women’s empowerment on campus; working with various administrators (i.e., program leaders, residential staff, etc.); and/or knowledge of existing resources on campus to begin developing feasible action plans. Identified solutions and strategies are summarized in Table 2.

Participants felt most strongly about pursuing the two identified actions to address their concerns around male-dominated social events: (1) incentivizing non-Greek, mixed-gender houses to throw parties to offer a socializing alternative to Greek parties; and (2) creating parties and events thrown by women. Decision-makers to target as potential collaborators to help implement these solutions included the University’s Residential Education office, student-run housing and Greek councils, and the University-run Women’s Center.

## 4. Discussion

### 4.1. Lessons Learned

#### 4.1.1. Data Collection Process

Previous *Our Voice* projects most often involved participants collecting data in pairs during scheduled walks within their community, in part for safety reasons. During our 2017 methodological test, we found that the scheduled walk method was problematic in the SV context because it potentially biased results by directing participants to concerns about outdoor assaults from strangers rather than other, more common, instances of campus SV. We determined that collection of community-level SV risk factors would be better suited for a more fluid data collection method, in which participants collect data throughout their normal day over a multiple-day period. Additionally, although community facilitators encouraged participants to collect data on specific experiences during the multi-day period, some data relied upon recollections and generalizations. This suggests that in future applications of the *Our Voice* model it will be important to give participants more training on this subject.

#### 4.1.2. Priority Setting/Action

As shown in Table 1, participants identified Greek-life as a source of perceived danger, echoing sociologist Elizabeth Armstrong’s work, which highlights how fraternity-run parties, where men control key aspects of the social life, foster SV [59,60]. Other studies have similarly identified Greek life as creating an increased risk of campus SV [61]. Yet participants did not prioritize Greek life or fraternities as an area for direct action. Instead, the participants focused on displacement strategies like boosting social activities outside of Greek houses, particularly with an emphasis on women-directed parties. It is unclear why participants chose indirect methods of addressing Greek life issues. It may be that participants prioritized what they saw as easier targets of opportunity. It may be that the gender dynamics of campus life, which privilege the needs of high-status males, including members of fraternities, influenced female student participants to choose other targets for reform. We conjecture that the priority setting and action steps are where SV researchers will want to be thoughtful in eliciting participants’ thinking and expectations. The participants are thinking through several dynamics at once: the relative impact of these issues on themselves and the community, the tractability of solutions, and implementation in the hierarchical power structures that exist on campuses. In future iterations of the *Our Voice* model applied to SV, a debrief of the priority setting process will be useful to better understand why some proposals received lower prioritization or were abandoned and to help address barriers that prevent students from pursuing specific issues. In particular, we are concerned that participants are likely to limit their range of issues and solutions in anticipation of harsh social and political reactions to their objectives. Researchers looking to deploy *Our Voice* in the SV setting on campuses should be thoughtful about students concerns of interpersonal and institutional betrayal [62,63].

### 4.2. Limitations

As noted in Section 2.2, in this study student facilitators chose to target exclusively self-identifying women students, and demographic information was not collected in detail. Although within the SV research field single-gender groups are common, this is a limitation of this particular instantiation of the method [22]. Depending on the research question, it is likely that researchers will want to bring together several different groups, of various composition, of members of the community. This is an important aspect of the model—it is likely that varying group composition will lead to variation in data collection, identification and prioritization of problems, and related solutions. Understanding the variation in how these different groups reach consensus is likely to lead to a better sense of the range of challenges and potential solutions available in the target community. We recommend future studies applying the *Our Voice* model consider forming multiple groups of different types of community members—including groups that are representationally cross sectional and groups that represent mixtures of subgroups (e.g., single gender groups, groups that are intentionally mixed with respect to political ideologies).

### 4.3. Looking Forward

Campus-based SV prevention strategies and programs suffer from a gap in knowledge of modifiable factors at the campus community level. We piloted the use of the *Our Voice* citizen science model as a means for generating both contextually-meaningful data and relevant, actionable solutions that could be taken by community members from a university. This pilot study focused on tailoring and implementing the first three steps of the 4-step *Our Voice* model within the SV context: the planning and recruitment step, in which relevant questions and group structures were formed to best target the broad community-level factors of interest; the data collection step, during which students collected relevant data using the SDT mobile app to identify places and situations on campus of relevance to gender-based issues and security; and the priority-setting step, during which students met together in a facilitated process to prioritize issues and brainstorm potentially relevant and feasible solutions for mitigating SV risk. As the school year ended, time constraints prevented the implementation of the remaining group action step, in which *Our Voice* participants typically present their findings to community decision-makers and stakeholders, discuss relevant solutions, and work together to enact the solutions identified.

While future plans involve the implementation and evaluation of all four steps of the *Our Voice* model, the application of the group action step in the campus SV context requires further consideration and exploration. Unlike many of the health-improvement collaborations between community members and stakeholders fostered through previous *Our Voice* studies, it is possible that student community members may experience a stronger pushback from campus stakeholders. Institutions tend to perceive both legal (e.g., Federal Title IX laws) and reputational risks associated with addressing sexual violence on campus. Concerns about these risks can lead administrators and other stakeholders to attempt to co-opt efforts that draw attention to these issues on the campus [19,62]. Additionally, as discussed above, power dynamics that are gendered, racialized, or divide along economic lines, may affect the priority assigned to particular solutions. The implementation of the group action step may need additional planning and information brought into the process to facilitate effective communication with administrators. It is possible that certain SV interventions may be constrained by law and researchers need to be prepared to offer guidance on these issues if they arise—e.g., if the group action conflicts with state laws.

### 4.4. Useful Features of the Our Voice Model for Developing Interventions to Reduce SV

As a research model, *Our Voice* represents a novel approach by including the following suite of features: (a) the participants determine what interventional steps are taken; (b) the community-engaged assessment process generates data from the community itself (e.g., photos and geospatial data), which are different from the types of quantitative surveys often obtained in epidemiological and intervention research and—given the contemporaneous collection by the participants themselves—reflect arguably more ecologically valid and meaningful data to community members themselves; (c) it offers the researchers insights into the decision-making process of community members as they identify and prioritize issues, and also as they discuss potential solutions; (d) it produces several potential interventions that community members identify as desirable; and (e) after selecting interventions, it scaffolds a process for participants to advocate for and implement interventions generated and conducted by the community itself. Running several iterations of the program—with intentionally different groups of students and with intentionally different guiding questions—would lead to useful insights about how a campus culture can be shifted to help prevent SV.

Within this pilot study, student participants used the SDT app to collect data on community factors related to their feelings of safety and risk of SV while moving normally through their community. This approach differs from SV research that has often used campus climate surveys to learn when students feel at risk of SV. While a useful source for insights, that type of survey lends itself to recall bias. In contrast, by applying the *Our Voice* method, students collect data on environments and settings where they feel at risk in situ and note their feelings in the moment. Conversely, some advantages of climate surveys are that they are anonymous and the data is easily comparable across institutions and time. We argue that both approaches have value and should be used in concert to improve the quality of available data.

College campuses are beginning to implement evidence-based SV prevention programming. Yet, to date, these strategies are focused on individual and interpersonal change with a noticeable absence of community-based organizing and intervention. The *Our Voice* model is a useful engine for generating creative, community-responsive interventions that may lead to new insights in the research and campus communities. One can imagine how a multi-site deployment of the *Our Voice* model (e.g., across several universities) could lead to a better understanding of how SV prevention can be both community-specific and reflective of more generalized dynamics shared across diverse communities.

## 5. Conclusions

The *Our Voice* model encourages groups of students to collect local data about SV, generate actionable solutions to reduce SV risk on campus, and to develop intervention strategies for factors that they identify as fostering an SV-risky environment. This empowers students to co-create a safer environment for all members of their community, and to contribute to reducing rates of SV on campus. In addition to the progress made by acting on these solutions, cycling through the *Our Voice* process with different groups allows researchers to understand how different group-level ideas emerge, and which members of the community are more or less activated by certain issues and solutions. Themes that are common to different groups can provide researchers with insights into community-level factors that can be prioritized in further research.

Our pilot work suggests that the *Our Voice* process can be deployed as a method to complement current SV prevention strategies implemented at universities. Specifically, *Our Voice* offers a relevant method to both understand community concerns and identify opportunities for developing a more comprehensive approach for addressing the high rates of SV on college campuses. By including programming targeted at the multiple levels of the social ecological model, campuses are more likely to reach the ultimate end goal—a reduction in attempted and completed SV.

## Figures and Tables

**Table 1 ijerph-18-07933-t001:** Examples of data collected in pilot study by theme.

Theme	Example Photo	Example Quotes
Greek life (n = 10)	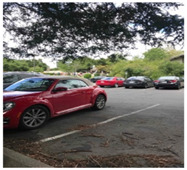	“I think the frats near my house are going to be partying all day, which is frustrating & makes me anxious. The amount of airwaves they take up on campus is infuriating.” (−)
		“I went to [a fraternity house on campus] and they had a consent sign and ID check [out] front. I appreciated these signs and felt safer knowing only Stanford students were allowed to enter.” (+)
Dorm culture/structure (n = 9)	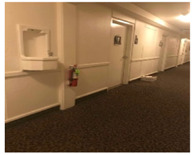	“Since both restrooms on [my] floor [are] gender neutral there have sometimes been situations where myself or another female resident will walk in and a male resident will leave the stall open treating [it] as a pseudo urinal and that can be a little uncomfortable.” (−)
		“I love my pod in my house, it’s all womxn and one gay man and so sharing a bathroom feels very comfortable to me.” (+)
Academic/professional/organizationalculture (n = 8)	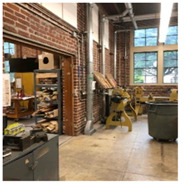	“The [design lab] is normally not intimidating, but today there was only men in the woodshop, who didn’t acknowledge my presence, but would talk shop to each other. I felt left out because of my gender.” (−)
		“My major and our classes are very gender balanced, and…the girls in our class feel comfortable interjecting and asking questions. Our professor in this course is a reputable and intelligent woman, which is what made me start reflecting on this.” (+)
Large social events (n = 4)	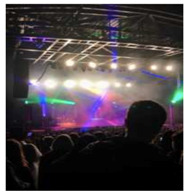	“There was also a really strong feminine presence at Frost [a university-sponsored music festival]—lots of women cheering” (+)
		“At Frost on Saturday, I was keeping check for safety & not only feeling physically small as a woman but also just I was surrounded w/men during the concert and felt this a lot.” (−)
Gyms (n = 3)	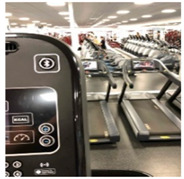	“At the gym I often feel my gender a lot—both in terms of body judgement and also in terms of what [equipment] I use to workout.” (−)
		“The way people look at each other at the gym makes me very conscious of my body and my gender. An old man stood behind a girl on the elliptical and was looking at her butt.” (−)

*Notes: *1. This is not an exhaustive list of themes, but the most common and most discussed themes. 2. The plus (+) and minus (−) symbols indicate whether the statement is a positive or negative concern. 3. To keep anonymity, we did not include a Greek life photo of a specific fraternity and instead used another photo that was submitted in regards to Greek life.

**Table 2 ijerph-18-07933-t002:** Major themes and solutions determined by community members in pilot study.

Theme	Reasoning	Potential Solution(s)
Dorm bathrooms	Male students using stalls without closing/locking door Place where participants felt very aware of their gender	Increasing access to all-gender/single-stall restrooms Making stalls more private (increasing door height, etc.) Discussions within dorms about bathroom use norms
Campus cultural norms	Perceived lack of regard among male students for how their behavior affects others	Including male speakers in freshmen orientation events to share experiences with healthy masculinity
Social events	Majority of all-campus parties are hosted by fraternities/male-dominated spaces Few non-Greek/non-alcoholic social alternatives	Increasing campus funding for non-Greek spaces to host parties as alternative social options Allowing sororities to host all-campus parties
Academic gender norms	Male instructors/students “speaking down” to or speaking over non-male students Need for increased female/nonbinary academic mentors	Including question on gender dynamics in course evaluation forms (e.g., “Did you feel respected/comfortable asking instructors for help,” “Did you feel that your voice was heard in this class?”) Hiring more female professors/TAs

## Data Availability

Not applicable.

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
