# Peer review of "A Novel Model for Generating Creative, Community-Responsive Interventions to Reduce Gender-Based Violence on College Campuses"

_ijerph, 2021, doi:10.3390/ijerph18157933_

Round 1

Reviewer 1 Report

Dear authors,

than you for this interesting manuscript. I have enjoyed it. However, several adjustments are needed before the article can be published. I have prepared a list of them for you:

  • the purpose of the article must be clearly stated in the introduction
  • the "introduction" section should be more structured and the authors should accompany the reader more with the text
  • the section describing data analysis should be significantly expanded, as this section points to the validity of the data
  • I would appreciate the addition of a section describing the limits of research and the ethical principles applied in the implementation of research

Reviewer 2 Report

The authors have chosen to address an important topic - community level prevention of sexual violence on college campuses. The citizen science approach and use of a data collection app are innovative and an interesting method for the campus sexual assault field.

Introduction - on balance spends a good deal of time on risk factors for SV when the focus is on prevention. The paper would be more effective if it centered prevention and theories/models of campus prevention strategies more centrally. There was notably some work related to community level campus work that was missing including: Potter et al's work on social marketing Know Your Power and McMahon, S. (2015) article on community level factors related to bystander intervention. McMahon, S. (2015). Call for research on bystander intervention to prevent sexual violence: The role of campus environments. American journal of community psychology55(3), 472-489. and also McMahon, S., Steiner, J. J., Snyder, S., & Banyard, V. L. (2019). Comprehensive prevention of campus sexual violence: expanding who is invited to the table. Trauma, Violence, & Abuse, 1524838019883275.  The frameworks in these articles go beyond the social ecological model and might be a helpful prevention framework for the paper. I also think some of the work on including youth voices in prevention is relevant here: Edwards, K. M., Jones, L. M., Mitchell, K. J., Hagler, M. A., & Roberts, L. T. (2016). Building on youth’s strengths: A call to include adolescents in developing, implementing, and evaluating violence prevention programs. Psychology of violence6(1), 15.

The state of SV prevention also calls for engagement with theories of intersectionality. Indeed the authors mention the web of prevention but notably absent is engagement with the ways that prevention across a community may need to do different things to reach different constituents. See for example: McMahon, S., Burnham, J., & Banyard, V. L. (2020). Bystander intervention as a prevention strategy for campus sexual violence: perceptions of historically minoritized college students. Prevention science21(6), 795-806. I think the method the authors present would be particularly helpful in this context - for engaging parts of college campuses who have historically been marginalized and silenced and left out of prevention planning.

The use of a citizen science model is interesting - it would be helpful if the authors described it a bit more generally first including how it is different from community based participatory action research, for example, before discussing Our Voice specifically. Readers need to understand more fully how this method will give us new information or create better prevention. Currently the authors compare Our Voice to ethnography and focus groups but PAR and specifically CBPR and methods like photovoice (which all have been used related to sexual violence) seem a better comparison.

The sample seems quite narrow and self selected. How might this have biased the view of sexual assault that was generated by the project?

The discussion highlights the potential uses of the method, limitations, and changes that might be made. But overall I think the discussion needs to discuss how this method produced findings that are really different from what we already know about sexual assault prevention. On balance I wanted more of an analysis of what we learn that is really different by using this method.

Reviewer 3 Report

The purpose of this article is to arise interest in models of self-prevention about Sexual Violence (SV). The authors propose a model—called Our Voice— which involves the same students in US campuses to detect factors are related to perpetration, victimization, and by-standers behaviors related to SV. See abstract lines 4-6.

Principally, this model generates data from the experience of the community itself, and produces several potential interventions that community members identify as meaningful. Then, the authors’ model focuses on giving ownership of the issue and its solutions to the community of potential victims, ensuring this way initiatives are tailored for a community and are more likely to trigger effective change at all levels (See: Cohen, L., Chávez, V. & Chehimi, S. (Eds.) (2007). Prevention is primary: Strategies for community well-being. San Francisco: John Wiley & Sons, Inc., partly available in Morgan J. Curtis, Engaging Communities in Sexual Violence Prevention. A Guidebook for Individuals and Organizations Engaging in Collaborative Prevention Work, available online: http://taasa.org/wp-content/uploads/2014/10/Engaging-Communities-in-Sexual-Violence-Prevention.pdf).

Paragraph 1. Introduction provides basic statistics especially by the side of CDC, the Center for Disease Control and Prevention, on the SV in the United States.

The statistics authors take into consideration are especially targeted on the SV on US campuses, where the presence of certain ethnic populations’ components—such as the Greek one—as well as their specific behaviors on campuses can increase the likelihood for SV. 

A 2019 Association of American Universities survey on sexual assault and misconduct polled over 150,000 students at 27 universities. Data revealed a 13% nonconsensual sexual conduct rate. Statistics provided by the Rape, Abuse, and Incest National Network also indicate that female college students between 18-24 remain three times more likely to experience sexual violence. Young adults ages 18-34 are at the highest risk and represent 54% of sexual assault cases. One out of every six women falls victim to completed or attempted sexual assault within their lifetime. Furthermore, women ages 18-24 not attending college face a 20% higher risk of falling victim to sexual assault. While sexual violence remains a concern, the number of cases has fallen 63% since 1993. See online: https://www.bestcolleges.com/resources/sexual-assault-on-campus/.

The authors embrace a multifaceted intersectional ecological model may explore the complex interplay between individual, interpersonal, community, and societal factors that contribute to SV experiencing.

In their paragraph 1.1. The Need for Community-Level SV Prevention Programming, the authors highlight campus-wide SV prevention programming. These are generally focused on individual-level and interpersonal-level factors. Instead the authors propose an ecological model at community- and environmental-level factors, which can include different risks for SV are extended not only to campus policies, features, and cultures, but also to a general environmental atmosphere there could be potentially SV-prone.

A community-level model is also likely to affect interpersonal interactions without necessitating case-by-case interventions, changing behaviors without intervening on individuals.

This attempted fostering change within a community requires the recruitment, activation, and continual engagement of community stakeholders, and so doing it grants to identify specific targets or change are particularly relevant to that community.

The authors quote of a program implemented in Kampala, Uganda, entitled SASA!, has demonstrated that community-level program was associated with a reduction in rates of SV.

The authors make reference also to a successful program is aside from SV, but it implies similar techniques related to the root causes of community violence in eight underserved US communities.

In paragraph 1.2. Introduction to Citizen Science – The Our Voice Initiative, the authors enter their own model Our Voice. The model levers on community facilitators, such as community leaders, researchers, or representatives, which use flyers, email, word of mouth, and/or other relevant strategies to recruit participants for the collection, analysis, and use of their own data to advocate for meaningful and sustainable change within their community.

The participant collect data using the Stanford Discovery Tool (SDT), a smartphone-based app available on both Apple and Android platforms. Through the app the participants can geo-tag photos and record audio or text narratives about the local environmental factors that can positively or negatively impact the SV on US campuses.

Next paragraph “Our Voice: Comparison with Existing Methods” is an explanation of the qualitative research the authors mean to diffuse among the scientific community. The authors’ model is similar to Focus Group’s method, where however the participants do not use of the SDT app and just discuss about their experiences and feelings on a specific topic.

In addition, respect to Focus Groups, an SDT Data Portal is available to the same participants to be coded and organized on the basis of their own experiences in the community.

Next paragraph “Our Voice: Previous Work”, is related to the application of SV-community-environmental prevention also for the minority group of LGBT (Lesbian, Gay, Bisexual, Trans) People.

In section 2. Materials and Methods, the authors specify Our Voice model targets especially SV on college campuses in the US and abroad. The study took place at Stanford University located on the west coast of the United States (after a pilot test conducted there in Spring 2019) and focused specifically on undergraduate women.

Four crucial questions are asked to participant at page 7 of the paper, all revolving around social spaces or situations at the university where the individuals may feel themselves uncomfortable, unsecure, and/or uneasy.

Paragraph 2.1. Selecting the Target Population explains how participants in the pilot study were recruited through emails and mailing lists.

Paragraph 2.2. Data Collection Process explains how participants have been taught to download and sign into the SDT mobile app with the aim at capturing photo and audio or text narratives about their feelings or experiences of risk and safety at the university. The method for collecting personal data by participants, and not secondary data heard or experienced by others, is detailed described at page 8 in the paper.

In Section 3. Results, the authors show data for 18 undergraduate students responded to the initial email has served as call for action about SV community-level prevention in the study. Other 10 undergraduate students were enrolled in the pilot study. Race/ethnicity of participants are enlisted including LGBT people.

The SDT mobile app has enjoyed of photos and text/audio narratives on a total of 102 walks in campus, and all materials have been managed by the research team as well as by the student facilitators.

The second of the two sub-paragraphs at Section 3. Priority Setting, it shows themes identified by participants. The authors notwithstanding the encouragement to provide for direct experience materials by participants to the study, have found some data use recollections and generalizations.

Table 1. at page 9 in the paper shows examples of data collected in pilot study by theme.

Next paragraph Solution Building shows the prioritization in the collected materials of four major themes, such as dorm bathrooms, campus cultural norms around male behavior, and social spaces/events.

Table 2. collects major themes and solutions determined by community members in pilot study.

Section 4. Discussion sets forth Our Voice models’ method. It focuses on aspects of Greek life may create an increased risk of campus SV, as it had been already explained in paragraph 1. Introduction. It seems to the authors Greek life issues on SV are determined by the openness of this ethnic component in campus colleges, and/or by their behaviors in social life.   

Next paragraph Looking Forward is a perspective paragraph on the interest an alike mobile app such as SDT proposed by authors may be able to identify places and situations on campus of relevance to gender-based issues and security.

The issue on SV on campus colleges is spiky, since it involves many sectors of the Society at large putting at question gender roles, racial features, or dividing economic lines in the US. It seems alike the same administrators would like to free themselves off from this cumbersome duty, because it often conflicts even with state laws.

So that the implementation of community-environmental level evidence-based SV prevention is therefore welcomed, since it enhances communities to take the role of “better hypothesis” good administrators of their own campus colleges lives.

Section 5. Conclusions is a reiteration of the need to deploy a community-level SV-risky environment analysis.

Well, the paper is interesting, and it demonstrates the authors have a high concern for the Sexual Violence (SV) prevention on US campus colleges. It is evident, someone among authors has had a direct experience on it, and I have appreciated the paper’s background on an “unexpressed fear” is somehow hidden in the paper, though also lively perceptible.

This told, I sincerely have not understood these are the outcomes of the pilot study in 2019, or those of a subsequent inquiry. I have not also understood whether you wish to do a beautiful work for IERPH MDPI, or else wish yourself just to publish here a scrap of your research, and then publish a more substantial and effective paper in another Journal.

In addition, Our Voice is currently the movement in favour of Julian Assange, and other so-called martyrs of the international justice.

MAJOR CHANGE REQUEST:

  1. I think you should change the name of your own model, if this must be applied to the SV-risk prevention on US campus colleges.

You can eventually call it Our Voice Against Violence on Campuses, or Our Voice Against Sexual Violence on Campuses, I personally think.

  1. If I were you, I would state clearly which Campus you have targeted for your research.

Should you be able to avoid the impression you are asking for cues for incriminating a specific Campus in the US, I think you should be able to obtain the permission.

I am available to help you on this, should you ask to me.

Stanford is a beautiful Academic Institution, and we should not encounter matter on it.

Problems with SV on University Campuses are often as those regarding the Army: there, it is probably easier to encounter such issues, if the H.R. 2527 has become a bill that extends a United States Department of Veterans Affairs (VA) program of counseling and care and services for veterans for sexual trauma that occurred during active duty or active duty for training to veterans who experienced such trauma during inactive duty training.

We cannot incriminate the Army, though we can try to find solutions to render “bad customs” out of interest by the side of communities.

  1. Please shorten your paragraphs and focus earlier on the search for materials collected by your SDT’s participants and related to photos and record audio or text narratives about the local environmental factors that can positively or negatively impact the SV on US campuses.
  1. Please avoid redundant paragraphs explain your general qualitative method on SDT mobile app, and instead describe—at least in one paragraph—fundamentals on CPTED – the Crime Prevention through Environmental Design.

It seems to me your Table 1.—with some pictures more, say 10—would have more interest to the reader, explaining previously which environmental factors are thought to instill fear of criminal victimization on a College Campus (see, for instance: Steinmetz, N.M.; Austin, D.M. Fear of Criminal Victimization on a College Campus: A Visual and Survey Analysis of Location and Demographic Factors. Am J Crim Just 39, 511–537 (2014). https://doi.org/10.1007/s12103-013-9227-1). There is plentiful literature on the topic!

Also, the Bureau of Justice Statistics has collected from a large sample of agencies throughout the USA, factors are related to applied forms of crime prevention, including crime prevention through environmental design, problem-oriented policing, and participation in anti-fear campaigns (see: Reyns, BW.; Henson, B. Crime prevention on college campuses: correlates of problem-solving, environmental design, and anti-fear efforts by campus law enforcement. Crime Prevention and Community Safety 23, 69–86 (2021). https://doi.org/10.1057/s41300-020-00108-4).

Through your positive/negative concerns in statement in the collected materials you could also easily add a simple frequency ratio Table and/or Graph of them, to give a more meaningful sense to your collected pictures, audio, and recorded narratives.

MINOR CHANGE REQUEST:

  • Please clear at the start in your abstract that the main method for Our Voice model on detecting Sexual Violence (SV) in US campus colleges is the SDT (Standard Discovery Tool) mobile app with the aim at capturing photo and audio or text narratives about the participants’ feelings or experiences of risk and safety at the university;
  • Please solve the acronym for CDC (Center for Disease Control and Prevention) at the start of paragraph 1. Introduction;
  • Please convert the text in footnote no. 1 at page 2 in the main body between parentheses;
  • Please add an acronym to Standard Discovery Tool (SDT) mobile app, quoted first time at page 5, and thereafter use SDT to mention it;
  • Please specify whether the photos and record audio or text narratives about the local environmental factors that can positively or negatively impact the SV on US campuses, or SV in general;
  • Please solve the acronym LGBT first time appears in your paragraph “Our Voice: Previous Work” at page 6;
  • Please specify whether Our Voice model targets especially SV on college campuses in the US and/or abroad, or else both.

Kind Regards,

Round 2

Reviewer 2 Report

The manuscript has been strengthened and improved. I have just some minor further suggestions. 

The term citizen science should appear in the first paragraphs of the paper. Our Voice is referenced but not identified as citizen science. Instead CBPR is referenced, which is good, but specific reference to citizen science is also needed in opening of the paper.

I appreciate the added sections to describe community-level interventions and added citations. I think that section should conclude with some sort of statement about how methods of citizen science can address the needs described in those paragraphs. As it stands it takes a long time for the reader to be introduced to the term citizen science.

Overall the paper could add a few sentences to define community level factors and community level interventions. These terms are used frequently but not really well defined initially in the paper.

Demographics of participants should be more specific. 

Authors indicate that in the discussion they note issues related to sample selection but I didn't see that new description in there. Indeed a clear section on limitations of the current study would be helpful.

Reviewer 3 Report

Dear Authors,

I thank you very much for your detailed replies to my review.

The paper has been strongly improved. It demonstrates a maturer style and a clearer exposition of both existing framework on Sexual Violence's (SV) reduction through community-based partecipatory research and your intent in editing such issues.

The outcome is more interesting to the reader than before, and I think it is now a pleasure to read it.

With Kind Regards,
